# *Perilla frutescens* Leaf Alters the Rumen Microbial Community of Lactating Dairy Cows

**DOI:** 10.3390/microorganisms7110562

**Published:** 2019-11-13

**Authors:** Zhiqiang Sun, Zhu Yu, Bing Wang

**Affiliations:** College of Grass Science and Technology, China Agricultural University, Beijing 100193, China; szq141835@sina.com (Z.S.); yuzhu33150@sina.com (Z.Y.)

**Keywords:** 16S rRNA gene, cow, *Perilla frutescens*, rumen microbiota

## Abstract

*Perilla frutescens* (L.) Britt., an annual herbaceous plant, has antibacterial, anti-inflammation, and antioxidant properties. To understand the effects of *P. frutescens* leaf on the ruminal microbial ecology of cattle, Illumina MiSeq 16S rRNA sequencing technology was used. Fourteen cows were used in a randomized complete block design trial. Two diets were fed to these cattle: a control diet (CON); and CON supplemented with 300 g/d *P. frutescens* leaf (PFL) per cow. Ruminal fluid was sampled at the end of the experiment for microbial DNA extraction. Overall, our findings revealed that supplementation with PFL could increase ruminal fluid pH value. The ruminal bacterial community of cattle was dominated by Bacteroidetes, Firmicutes, and Proteobacteria. The addition of PFL had a positive effect on Firmicutes, Actinobacteria, and Spirochaetes, but had no effect on Bacteroidetes and Proteobacteria compared with the CON. The supplementation with PFL significantly increased the abundance of *Marvinbryantia*, *Acetitomaculum*, *Ruminococcus*
*gauvreauii*, *Eubacterium*
*coprostanoligenes*, *Selenomonas_1*, *Pseudoscardovia*, *norank_f__Muribaculaceae*, and *Sharpea*, and decreased the abundance of *Treponema_2* compared to CON. *Eubacterium coprostanoligenes*, and *norank_f__Muribaculaceae* were positively correlated with ruminal pH value. It was found that *norank_f__Muribaculaceae* and *Acetitomaculum* were positively correlated with milk yield, indicating that these different genera are PFL associated bacteria. This study suggests that PFL supplementation could increase the ruminal pH value and induce shifts in the ruminal bacterial composition.

## 1. Introduction

Plant secondary metabolites are naturally produced by plants or special medicinal herbs and can be readily formulated into feed rations [1]. The bioactive compounds derived from medicinal herbs have been discussed as potential alternatives to conventional antibiotics in ruminant production [2]. *Perilla frutescens* leaf (PFL) is a commonly used medicinal herb with antibacterial, antiallergy, anti-inflammatory, antioxidant, antidepressant, and anticancer properties due to its secondary metabolites including phenolic compounds, flavonoids, triterpenes, anthocyanin, and α-linolenic acid [3,4]. The use of medicinal herbs and plant secondary metabolites in livestock has been a hot topic in veterinary research and practice [5], regarding the increasing concerns and challenges of using antimicrobials in livestock production and the proliferation of resistant bacteria via the food chain [6].

It has been established that the rumen microbial community has a profound impact on the performance, health, and immunity of the host animal [7,8]. Previous studies found that the effect of ruminal microbiota on the host can be achieved by the short chain fatty acids released by ruminal bacteria such as *Mitsokella spp* [9]. In addition, Shen et al. [10] found that the effect of rumen-derived short chain fatty acids on the growth and metabolism of epithelial cells was mediated by the regulatory network of G protein coupled receptor (GPR) and histone deacetylase (HDAC). Thus, the rumen microbiome has intimate connections with growth and development of the host animal. It has been found that the supplement of *Andrographis paniculata* leaves rich in plant active substances (lactones, flavonoids, sterols) in goat diet increased the quantity of ruminal *Ruminococcus albus*, *Ruminococcus flavefaciens*, and *Fibrobacter succinogenes*, and then improved the nutrients digestibility [11]. The extracts of *P. frutescens* exhibit intense antimicrobial activities [12] and mitigate methane emissions from ruminants [13]. Many previous studies found that plant secondary metabolites, such as phenols, flavonoids, and triterpenes, have strong regulatory effects on ruminal microorganisms [1,4,13,14].

To our knowledge, however, little research has been conducted to explore the effects of PFL on rumen fermentation and ruminal microbiota in dairy cows. In this study, we hypothesized that the PFL had effects on the rumen microbiota and rumen metabolism, possibly providing an efficient regulatory effect for the dairy cow feed industry. Therefore, this study was conducted to investigate the effects of dietary supplementation with PFL on rumen fermentation and ruminal bacterial community in dairy cows.

## 2. Materials and Methods

### 2.1. Animals, Diets, and Experimental Design

The use of the animals was approved by the Animal Care Committee of China Agricultural University (Beijing, China; approval no. AW26128102-1; approval date: 26 December 2018), and the experimental procedures used in this study were in accordance with the university’s guidelines for animal research. The animals used in this study were from a commercial dairy farm with approximately 1500 lactating dairy cows (Shandong Yinxiangweiye dairy farm, Shandong, China). The animals used in this study were housed separately with other cows using a specific subfield. Thirty-six Holstein dairy cows (initial characteristics: milk yield = 42.1 ± 8.70 kg/d; days in milk = 90.7 ± 35.0 d; parity = 3.0 ± 1.35; mean ± SD) were randomly divided into two treatment groups according to a randomized block design (*n* = 18). The treatment included a basal total mixed ration (TMR) diet without (CON group) or with dietary PFL at 300 g/d per cow (PFL group). The ingredients and chemical composition of the basal TMR are shown in Appendix A. The adaptation period was one week, and the trial period was 8 weeks. At the end of the feeding experiment, fourteen cows (each group had seven cows) were selected for the collection of rumen fluid based on the milk yield (close to the average milk yield of the group).

### 2.2. Sample Collection and Measurements

The PFL was chemically characterized with the following procedure. The PFL sample (50 mg) was weighed into an EP tube, and 1000 μL of extract solution (acetonitrile: methanol: water = 2:2:1) with 1 μg/mL internal standard (2-chloro-L-phenylalanine, purity ≥ 98%) was added. After 30 s of vortexing, the samples were homogenized at 35 Hz for 4 min and sonicated for 5 min on ice. The homogenization and sonication cycle was repeated 3 times. Then the samples were incubated for 1 h at −20 °C and centrifuged at 11,000 rpm for 15 min at 4 °C. The resulting supernatant was transferred to a fresh glass vial for LC/MS analysis. The detailed procedure of LC/MS analysis procedure was described in our previous study [1].

The individual milk yield was recorded every day with a milk-sampling device (Afkin Ranch Management Software, version 5.2, Kibbutz Afikim, Israel). During the experiment, a milk sample from one day (50 mL, 4:3:3, composite) was collected every two weeks from three milking events and added to the composite milk sample, which was then stored at 4 °C for future milk protein, fat, lactose, total solid, and urea nitrogen analyses.

Ruminal fluid was collected 3 h after the morning feeding on the last day of this experiment using an oral stomach tube according to Shen et al. [15]. The initial 150 mL was discarded. The fluid pH was measured immediately. The samples were immediately flash-frozen in liquid nitrogen and stored at −80 °C for DNA extraction, and another subsample was used to measure the volatile fatty acids (VFA) according to Hu et al. [16] and to determine the ammonia nitrogen according to Broderick and Kang [17].

### 2.3. DNA Extraction and Sequencing

Microbial DNA was extracted from rumen fluid (1.5 mL) samples using the E.Z.N.A. stool DNA kit (Omega Biotek, Norcross, GA, U.S.) according to the manufacturer’s protocols. The quality of DNA was evaluated by 1% agarose gel electrophoresis and spectrophotometry. The final DNA concentration and purification were determined by a NanoDrop 2000 UV–VIS spectrophotometer (Thermo Scientific, Wilmington, DE, USA). The 16S rRNA V3-V4 region of the prokaryotic ribosomal RNA gene was amplified using the following primers: 338F: ACTCCTACGGGAGGCAGCAG and 806R: GGACTACHVGGGTWTCTAAT, where the barcode is an eight-base sequence that is unique to each sample [18]. The DNA template was uniformly diluted to 10 ng for amplification. The PCR procedure was as follows: 95 °C for 3 min, followed by 27 cycles at 95 °C for 10 s, 55 °C for 30 s, and 72 °C for 45 s, and a final extension at 72 °C for 10 min; PCRs were performed in triplicate in a 20 μL mixture containing 4 μL of 5 × FastPfu Buffer, 2 μL of 2.5 mM dNTPs, 0.8 μL of each primer (5 μM), 0.4 μL of FastPfu Polymerase and 10 ng of template DNA. Amplicons were extracted from 2% agarose gels and purified using the AxyPrep DNA Gel Extraction Kit (Axygen Biosciences, Union City, CA, USA) and quantified using QuantiFluor™-ST (Promega, Madison, WI, USA) following the manufacturer’s protocol. Purified amplicons were pooled equimolarly and paired-end sequenced (2 × 300) on an Illumina MiSeq PE300 platform (Illumina, San Diego, CA, USA) following the standard protocols by Majorbio Bio-Pharm Technology Co. Ltd. (Shanghai, China). The raw sequences were deposited in the NCBI Sequence Read Archive (https://www.ncbi.nlm.nih.gov/sra/SRP222284) under the accession number SRP222284.

### 2.4. Bioinformatic Analysis

Raw fastq files were quality-filtered by Trimmomatic and merged by FLASH: the reads were truncated at any site receiving an average quality score < 20 over a 50 bp sliding window; sequences whose overlap was longer than 10 bp were merged according to their overlap with no more than 2 bp mismatch; sequences of each sample were separated according to barcodes (exact matches) and primers (allowing for two nucleotide mismatches). Reads containing ambiguous bases were removed. Operational taxonomic units (OTUs) were clustered with ≥ 97% similarity cutoff using UPARSE (version 7.1; http://drive5.com/uparse/) with a novel ‘greedy’ algorithm that performs chimera filtering and OTU clustering simultaneously. Venn analysis was performed to identify unique and common OTUs between the CON and PFL groups. The taxonomy of each 16S rRNA gene sequence was analyzed by RDP Classifier algorithm (http://rdp.cme.msu.edu/; Version 2.2) based on the SILVA database (https://www.arb-silva.de/). The confidence used for the classification is 0.7. Chao1, Simpson and all other alpha diversity indices were calculated in QIIME. The sequences were rarefied at the same value (25,965) for each sample.

The OTU rarefaction curve and rank-abundance curves were plotted in QIIME. A Kruskal–Wallis H test and Tukey’s HSD test were performed in R (Version 3.2.4, Auckland, New Zealand) to compare the alpha indices among groups. Multivariate statistical analyses, including a principal coordinate analysis (PCoA) were calculated and plotted in R (Version 3.2.4). The non-strict version of LEfSe was used to determine the bacteria most likely to explain the differences between PFL and CON by coupling a nonparametric factorial Kruskal–Wallis (KW) sum-rank test for statistical significance with additional tests assessing biological consistency and the relevance of effects. Bacteria with LDA scores greater than 3 were speculated to have a different abundance between the PFL and CON groups.

### 2.5. Correlation and Statistical Analysis

To explore the functional correlation between the rumen bacterial and rumen fermentation, and between rumen bacteria and milk synthesis, a Spearman’s rank correlation matrix was generated by calculating the Spearman’s correlation coefficient among the top 20 genera and candidate ruminal fermentation and milk performance variables in the R program, and only connections with a *p*-value of less than 0.05 and *r* > 0.54 were retained. The rumen fermentation characteristics were analyzed with a randomized complete block design using PROC MIXED of SAS (version 9.2, SAS Institute Inc., Cary, NC, USA). The results were reported as least squares means that were calculated and separated using the PDIFF option in SAS. Significance was declared at a *p*-value ≤ 0.05; trends were declared at 0.05 < *p*-value ≤ 0.10.

## 3. Results

### 3.1. Characteristics of PFL

We found 122 bioactive compounds in PFL using UHPLC-QTOF-MS (Appendix A). The top 10 enriched bioactive compounds were clareolide, betaine, sucrose, scutellarin, apigenin, L-valine, caffeic acid, 2-pyrrolidinecarboxylic acid, L-phenylalanine, and L-tryptophan. Most of these compounds are flavonoids, alkaloid, triterpenoids, amino acids, etc.

### 3.2. Milk Performance and Rumen Fermentation

Feed intake was not affected by diet, but it was affected by week (*p* < 0.01, Table 1). Milk yield was higher in the PFL group than in the CON group (*p* = 0.04), and was greater in cows fed PFL than in cows fed CON after 3 weeks of feeding (*p* < 0.01). Compared to the control group, the yield of milk protein (*p* = 0.04) and lactose (*p* = 0.01) were increased in the PFL group. A higher level of feed efficiency (milk yield/dry matter intake) was detected in the PFL cows than in the control cows (*p* < 0.01). No interactive effects between sampling time and treatment were observed for the abovementioned variables (*p* > 0.05).

Rumen pH was higher in PFL cows than in CON cows (Table 2). The molar proportion of the valerate concentration had a tendency to be higher in the PFL group than in the CON group (*p* = 0.08).

### 3.3. Change in Ruminal Bacterial Communities

The number of sequences (total and average) before filtering is 797,635 and 56,974, respectively. The number of sequences (total and average) after filtering is 455,756 and 32,554, respectively. There were 1858 OTUs that were identified in PFL and CON cows, among which 1670 OTUs were found in both groups and accounted for 90.0% of the total OTUs, indicating the presence of an extensive common microbiome (Figure 1A). Compared with the CON, the PFL group had more OTUs (*p* < 0.01; Figure 1A). The PCoA based on Bray–Curtis dissimilarity was applied to analyze the difference in bacterial composition between PFL and CON groups. The PCoA plots (Figure 1B) showed that the clouds derived from the PFL and CON data had a tendency to be separated from each other. A similar level of species richness existed between PFL and CON based on the Sobs, Shannon, Simpson, Ace, Chao, and coverage index analyses, which indicated that there was a similar tendency of diversity and uniformity between PFL and CON (Table 3).

In total, seven bacterial phyla were identified in the rumen samples that had relatively high abundances (>1%), including Bacteroidetes, Firmicutes, Proteobacteria, Patescibacteria, Spirochaetes, Tenericutes, and Actinobacteria (Figure 2A). There were 247 bacterial taxa identified at the genus level, and 27 genera were present with relatively high abundances (>1%, Figure 2B). For the genus level (Figure 2B), the percentage of unclassified sequences is 8.1% (20/247).

We identified 25 clades as PFL biomarkers, which distinguished the PFL group from the CON group (Figure 3). Twenty clades were more abundant in the PFL samples, including four genera belonging to the Clostridiales (*Marvinbryantia*, *Acetitomaculum*, *Ruminococcus gauvreauii*, *Eubacterium coprostanoligenes*), one genus belonging to the Veillonellaceae (*Selenomonas_1*), one genus belonging to the Bifidobacteriaceae (*Pseudoscardovia*), one genus belonging to Muribaculaceae (*norank_f__Muribaculaceae*), and *Sharpea*. In addition, at the order level, Clostridiales, Selenomonadales, and Bifidobacteriales were affected by PFL compared to the CON. At the class level, Clostridia, Negativicutes, and Actinobacteria were different between the PFL and CON. At the phylum level, Firmicutes and Actinobacteria were different between PFL and CON groups. Five clades were much less abundant in the PFL samples, including *Treponema_2* belonging to Spirochaetaceae at the family level, Spirochaetales at the order level, Spirochaetia at the class level, and Spirochaetes at the phylum level.

### 3.4. Correlation Analysis

It was found that the rumen fermentation indices were related to the ruminal bacteria community (Figure 4A). In detail, *Christensenellaceae_R-7* (*r* = 0.58, *p* < 0.05), *Eubacterium coprostanoligenes* (*r* = 0.62, *p* < 0.05), and *norank_f__Muribaculaceae* (*r* = 0.75, *p* < 0.01) were positively correlated with pH value. *Prevotellaceae_UCG-003* (r = 0.54, *p* < 0.05) and *Prevotella_1* (*r* = 0.56, *p* < 0.05) were positively correlated with ammonia nitrogen. *Prevotella_1* (*r* = 0.67, *p* < 0.01) and *unclassified_f__Prevotellaceae* (*r* = 0.60, *p* < 0.05) were positively correlated with acetate. *Prevotella_1* (r = 0.79, *p* < 0.01) and *unclassified_f__Prevotellaceae* (*r* = 0.76, *p* < 0.01) were positively correlated with butyrate. *Prevotella_1* (r = 0.68, *p* < 0.01), *Succiniclasticum* (r = 0.66, *p* < 0.05), *unclassified_f__Prevotellaceae* (*r* = 0.79, *p* < 0.01), and *Ruminococcus_2* (*r* = 0.62, *p* < 0.05) were positively correlated with valerate. *Prevotella_1* (*r* = 0.65, *p* < 0.05) and *unclassified_f__Prevotellaceae* (*r* = 0.56, *p* < 0.05) were positively correlated with total volatile fatty acids. *Rikenellaceae_RC9_gut* (*r* = 0.59, *p* < 0.05), *Prevotellaceae_UCG-001* (r = 0.67, *p* < 0.05), and *Prevotellaceae_UCG-003* (r = 0.76, *p* < 0.05) were positively correlated, but *Succinivibrionaceae_UCG-001* (*r* = −0.57, *p* < 0.05), *Prevotella_7* (*r* = −0.58, *p* < 0.05), and *Lachnospira* (r = −0.66, *p* < 0.05) were negatively correlated with acetate: propionate.

At the genus level, *norank_f__Muribaculaceae* (*r* = 0.68, *p* < 0.01) and *Acetitomaculum* (*r* = 0.59, *p* < 0.05) were positively correlated with milk yield (Figure 4B). *norank_f_Muribaculaceae* (*r* = 0.63, *p* < 0.05) was positively correlated but *Treponema_2* (*r* = −0.64, *p* < 0.05) was negatively correlated with energy corrected milk. *Eubacterium coprostanoligenes* was negatively correlated with milk fat content (r = 0.54, *p* < 0.05). *Prevotellaceae_UCG-003* (*r* = −0.57, *p* < 0.05) and *norank_f__F082* (*r* = −0.54, *p* < 0.05) were negatively correlated with milk lactose content. *Succinivibrionaceae_UCG-001* (*r* = −0.55, *p* < 0.05) was negatively correlated with milk urea nitrogen.

## 4. Discussion

The current study characterized the changes in milk performance and rumen bacterial composition and the linkage between these factors after cows were fed PFL. The PFL has been reported to mainly contain hydrophilic (phenolic compounds, flavonoids, and triterpenes) and hydrophobic compounds (volatile compounds, fatty acids, policosanols, tocopherols, and phytosterols) [4], which was similar to the results of our study. In our previous study, when dairy cows were fed triterpene saponins at a moderate dose, milk synthesis increased and immune function improved [19]. In addition, the ethanol extract of *P. frutescens* seeds used at a relatively low concentration decreased methane production without adversely affecting rumen fermentation [13]. Considering the characteristics of the herb used, we estimated that the increased milk yield and feed efficiency might be due to the beneficial role of the bioactive compounds of PFL in modifying rumen fermentation by inhibiting energy transfer to the methane production pathway.

Furthermore, we identified specific rumen bacteria that may contribute to rumen fermentation and milk performance. Our results revealed that the addition of PFL had a positive effect on Firmicutes, Actinobacteria, and Spirochaetes, but had no effects on Bacteroidetes and Proteobacteria phyla compared with CON. In addition, we found that the relative abundance of *Marvinbryantia*, *Acetitomaculum*, *Ruminococcus gauvreauii*, *Eubacterium coprostanoligenes*, *Selenomonas_1*, *Pseudoscardovia*, *norank_f__Muribaculaceae*, and *Sharpea* was higher, but that of *Treponema_2* was lower in the PFL than in the CON. In addition, *Acetitomaculum*, *Eubacterium coprostanoligenes*, *norank_f__Muribaculaceae*, and *Treponema_2* ranked among the top 20 genera and were highly correlated with milk performance. Thus, these four bacterial taxa may be associated with PFL.

The three dominant ruminal bacterial phyla in the current study were Bacteroidetes, Proteobacteria, and Firmicutes, which was similar to the findings of previous studies [20,21]. The phylum Firmicutes was increased after cows were fed PFL in the present study. There were close associations between Firmicutes and the degradation of structural polysaccharides [22]. We also found that the proportion of valerate in cows fed PFL tended to be greater than that in CON cows. Valerate can be produced by ruminal bacterial species of *Clostridium*, and the abundance of *Clostridium* was negatively associated with methane yield [23]. The class Clostridia and order Clostridiales belonging to Firmicutes were increased after cows were fed PFL. In addition, *Marvinbryantia*, *Acetitomaculum*, *Ruminococcus gauvreauii*, *Eubacterium coprostanoligenes*, *Selenomonas_1*, and *Sharpea*, belonging to Firmicutes, were increased after cows were fed PFL. Thus, we propose that the PFL might increase the ruminal degradation of structural polysaccharides accompanied by potential inhibition of methane production. However, the assumption should be confirmed by further research.

We found that the relative abundance of *norank_f__Muribaculaceae* was highly positively correlated with the ruminal pH value, which was significantly greater in cows fed PFL than in CON cows. The abundance of Muribaculaceae was an important predictor of short chain fatty acid concentrations in the guts of mice [24]. Metagenomics results suggested that populations of Muribaculaceae are equipped with fermentation pathways to produce succinate, acetate, and propionate [25] through the degradation of particular types of polysaccharides, including plant glycans, host glycans, and α-glucans [25,26]. Thus, Muribaculaceae has the ability to undergo acetogenesis, similar to *Acetitomaculum*, which belongs to the group of acetogenic bacteria, which are a diverse group of bacteria whose main unifying characteristics is the ability to synthesize acetate from H_2_ + CO_2_ [27]. As the abundance of *Acetitomaculum* increased in the current study, the competitive advantage of acetogenesis may have increased compared with that of methanogenesis, and this may reduce the energy lost as methane as well as the eructation of ruminal methane as a greenhouse gas [28]. In a previous study, the abundances of *Acetitomaculum* increased with increasing dietary energy [29]. *Acetitomaculum* mainly exists in ruminants fed a high-concentrate diet and can utilize monosaccharides to produce acetate [30]. In addition, the current study also found that *norank_f__Muribaculaceae* and *Acetitomaculum* were highly positively correlated with milk yield, indicating their potential role as bacterial biomarkers for milk synthesis. We also proposed that the cows fed PFL showed an increased energy allocation for milk synthesis via the increase in the abundance of ruminal *Muribaculaceae* and *Acetitomaculum*.

We found that the relative abundance of *Eubacterium coprostanoligenes* was increased by feeding cows PFL, which was also positively correlated with ruminal pH value, but negatively correlated with milk fat content. Our result was similar to a previous study that found that the abundances of *Eubacterium coprostanoligenes* were significantly higher in the high-yield dairy cow group than in the low yield group, and the high-yield had better ruminal fermentation patterns than the low yield cows [31]. In addition, our results revealed that the addition of PFL had a positive effect on pH value, which also showed that the increased pH value was associated with *Eubacterium coprostanoligenes*. To date, very few studies have focused the function of *Eubacterium coprostanoligenes*. Thus, the association between *Eubacterium coprostanoligenes* and milk fat and its relationship with pH requires further study. Quercetin did not prevent the growth of *Ruminococcus gauvreauii* [32]. In contrast, its aglycone quercetin exerted a strong inhibitory effect on *Ruminococcus gauvreauii* [33]. Naringenin strongly inhibited the growth of *Ruminococcus gauvreauii*, in the human gut [34]. In the present study, the PFL was found to be enriched with flavonoids, alkaloids, triterpenoids, etc., which was similar to the results of previous studies [3,4]. However, we found that the feeding of PFL to cows resulted in an increased abundance of *Ruminococcus gauvreauii*. To date, few studies have revealed the function of *Ruminococcus gauvreauii* [34]. We estimated that the increased abundance of *Ruminococcus gauvreauii* was due to the specific compounds in PFL, but this needs to be explored further.

In our study, an increased abundance of *Sharpea* in cows fed PFL was found. Bacteria belonging to *Sharpea* have been reported to utilize a wide variety of carbohydrates to produce lactate [35]. It was reported that the ruminal *Sharpea* of sheep produced low levels of methane [23]. A recent study found that the abundance of the genus *Sharpea* was much higher in the high milk yield and high milk protein content cows than in the low milk yield and low milk protein content cows [21], which was confirmed by the current study that the PFL cows with higher milk yield had greater abundance of ruminal *Sharpea*. In addition, *Selenomonas* that is also a lactate-utilizing bacteria [36] had greater abundance in PFL cows compared to CON cows. It was reported that *Selenomonas_1* abundance was positively correlated with C18:1n9t concentration, and *Selenomonas_1* participated in hydrogenation of long-chain fatty acids [29]. Thus, the increased abundance of *Selenomonas_1* as a result of feeding cows PFL might have had positive effects on milk fat synthesis in increasing the milk unsaturated fatty acid synthesis, but this speculation needs further validation via the characterization of the milk fatty acid composition and its relationship with ruminal bacteria. In addition, our finding of the increased abundance of *Selenomonas_1* in cows fed PFL was similar to a previous study that found that the extract of *Perilla frutescens* seeds effectively increased the abundance of *Selenomonas* [13]. We estimated that the cows fed PFL experienced a fermentation shift via the increased abundance of ruminal *Sharpea* and *Selenomonas_1*, thus contributing to higher milk yield [21]. However, this is just a speculation which should be confirmed by further studies for the functions of these bacteria in relating to milk performance.

A negative correlation was found between energy corrected milk and *Treponema_2*, and the relative abundance of *Treponema_2* was lower in PFL than in CON. It was reported that *Treponema* are cellulose-degrading microbes [37]. *Treponema* are a commonly detected bacterial group in the rumen that are involved in the degradation of soluble fibers [38]. However, a high-concentrate diet significantly decreased the abundance of *Treponema* compared with a high fiber diet [39]. The growth of the *Treponema* genus was remarkably supported by the inclusion of pectin [40]. However, to our knowledge, no previous study has found inhibitory effects on *Treponema* using PFL or its extract. Thus, the underlying mechanism of PFL in inhibiting *Treponema* should be revealed. It was only reported that several plant extracts had the inhibition roles of peptidase and giycosidase activities of *Treponema denticola* [41]. Based on the previous studies, we hypothesized that the inhibition of *Treponema* might be due to the specific bioactive compounds in PFL.

In addition to the different bacterial taxa between PFL and CON, *Succinivibrionaceae_UCG-001* and *Prevotella_1* were correlated with milk synthesis or rumen fermentation characteristics. *Succinivibrionaceae_UCG-001* comprised the core microbiome in young cattle [1], which was similar to the current study in adult cows. Members of the Succinivibrionaceae have been previously proposed to be responsible for the lower methane emissions in wallaby microbiota [42]. It was proposed that Succinivibrionaceae produce succinate which is converted to propionate by other members of the microbiota; thus, less hydrogen might be available for methanogens [43]. Thus, *Succinivibrionaceae_UCG-001* was negatively correlated with the ratio of acetate to propionate in the current study, which might have been due to the increased production of propionate in the rumen. In addition, its highly negative correlation with milk urea nitrogen might be attributed to the improved nitrogen utilization efficiency because of the energy saved by the decrease in methane production [44]. Our results showed that *Prevotella 1* had positive effects on VFA concentrations (acetate, propionate, and valerate), which was similar to a previous study that found a significant positive correlation between *Prevotella* and VFA [21]. *Prevotella* is a genus consisting of proteolytic, amylolytic, and hemicellulolytic bacteria dominating the rumen of adult dairy cows and producing succinate and acetate [45]. Therefore, the correlation analysis from the current study confirmed the function of *Succinivibrionaceae_UCG-001* and *Prevotella*.

## 5. Conclusions

In this study, PFL was found to be enriched with flavonoids, alkaloids, triterpenoids, amino acids, etc., and induced compositional shifts in the ruminal microbiota, as confirmed for the first time by high-throughput sequencing, indicating that PFL is a potential functional feed source or additive that can be used to regulate rumen fermentation. We identified several key differential bacteria that responded to PFL addition, such as *norank_f__Muribaculaceae* and *Acetitomaculum*, which were also correlated with milk yield. In the future, it may be possible to reveal the functional microbiome associated with PFL or its key secondary metabolites through metagenomics or meta-transcriptomics.

## Figures and Tables

**Figure 1 microorganisms-07-00562-f001:**
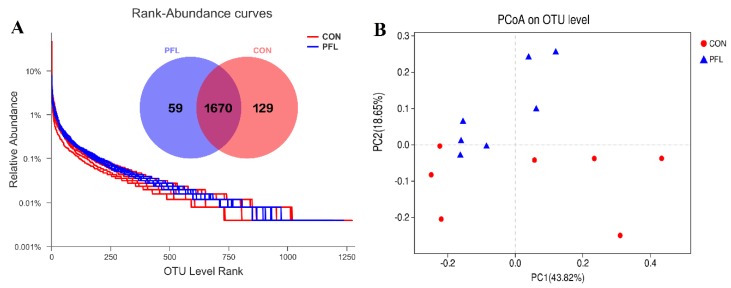
The rank-abundance curves derived from the OTU level. Venn diagram illustrating overlap of microbial operational taxonomic units (OTUs) at the 3% dissimilarity level among treatments (**A**). Unweighted principal coordinate analysis (PCoA) analysis of taxonomical classifications of rumen bacterial communities in the cows fed *P. frutescens* leaf (PFL) and control (CON) diets (**B**).

**Figure 2 microorganisms-07-00562-f002:**
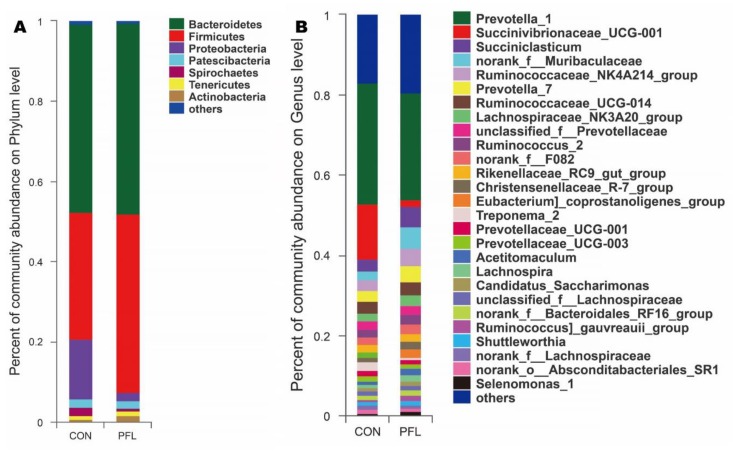
Distribution of bacterial taxa averaged under phyla (**A**) and genera (**B**) level across the dietary treatments (as a percentage of the total sequence).

**Figure 3 microorganisms-07-00562-f003:**
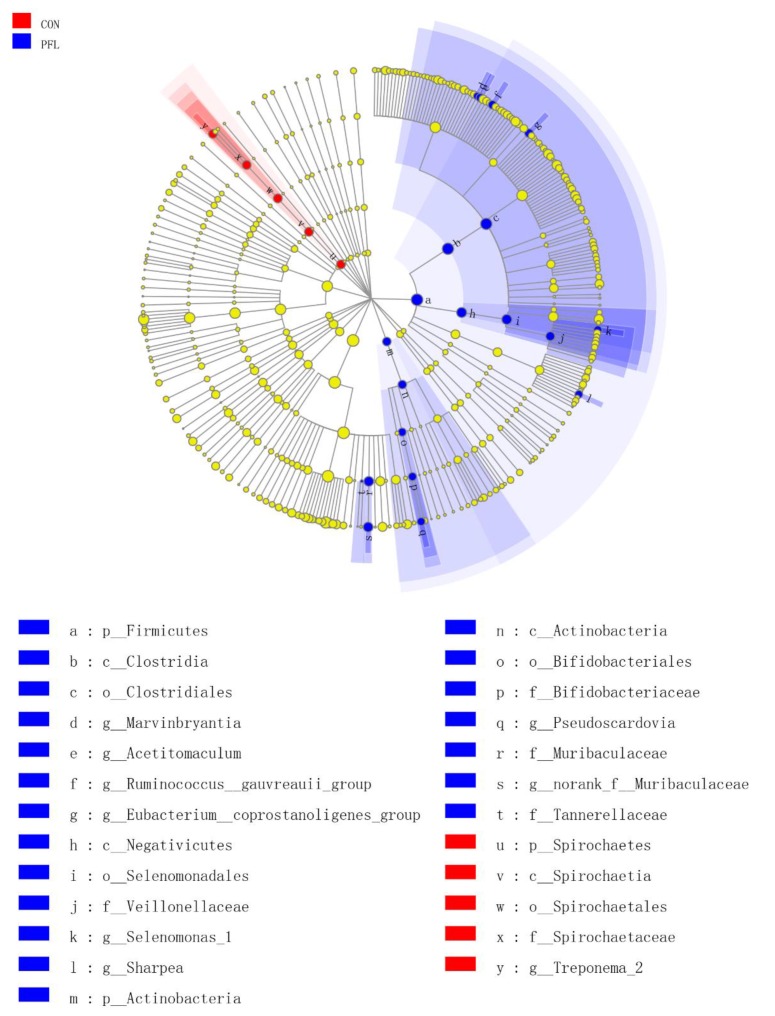
The ruminal bacteria (highlighted by small circles and by shading) showing different abundance values between *Perilla frutescens* leaf (PFL) and control (CON) groups. There are six layers from the inside of this plot to the outside, corresponding to six levels of taxonomy (kingdom, phylum, class, order, family, and genus). Each node (small circle) represents a taxon. Blue and red nodes represent the bacterial community of PFL with the significant higher and lower abundance in PFL compared with that in the CON group, respectively. Yellow nodes indicate the bacteria that are not statistically and biologically differentially between the two groups. The diameter of each circle is proportional to the taxon’s abundance.

**Figure 4 microorganisms-07-00562-f004:**
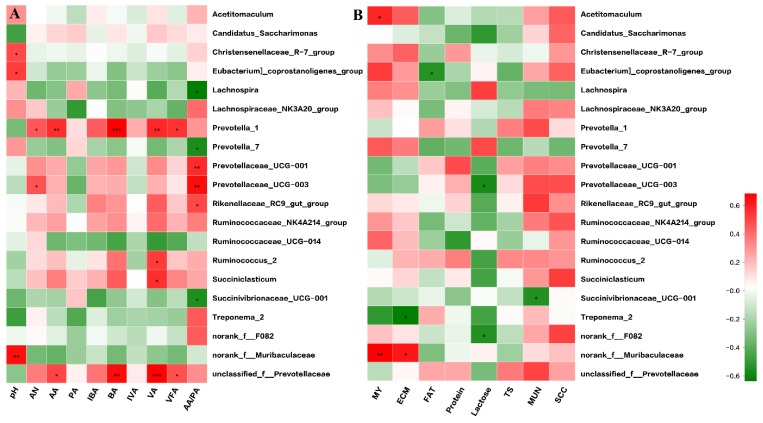
Correlation matrix between the top 20 ruminal bacterial genera and rumen fermentation parameters (**A**) and between the top 20 ruminal bacterial genera and milk performance variables (**B**). Positive correlations are shown in red and negative correlations are shown in green. The color intensity is proportional to the correlation values [r] within a correlation group. * *p* < 0.05; ** *p* < 0.01. AN = ammonia nitrogen, AA = acetate, PA = propionate, IBA = isobutyrate, BA = butyrate, VA = valerate, IVA = isovalerate, VFA = total volatile fatty acids, AA_PA = the ratio of acetate to propionate, MY = milk yield, ECM = energy corrected milk, TS = total solids, MUN = milk urea nitrogen, SCC = somatic cell count.

**Table 1 microorganisms-07-00562-t001:** Effect of *Perilla frutescens* leaf (PFL) supplementation on feed intake and lactation performance in dairy cows.

Item	CON	PFL	SEM	*p*-Value		
				Treatment	Week	Treatment × Week
Dry matter intake, kg/d	24.3	24.7	0.17	0.23	<0.01	0.63
Milk yield, kg/d						
Raw	36.8	39.2	0.77	0.04	<0.01	0.12
ECM ^a^	45.4	47.3	1.34	0.35	0.04	0.57
Protein	1.28	1.36	0.026	0.04	0.10	0.73
Fat	1.76	1.78	0.078	0.91	<0.01	0.41
Lactose	1.84	2.05	0.046	0.01	0.10	0.77
Total solids	4.98	5.27	0.127	0.12	0.07	0.66
Milk content, g/100g						
Protein	3.47	3.39	0.037	0.19	<0.01	0.58
Fat	4.80	4.40	0.187	0.15	<0.01	0.09
Lactose	5.05	5.10	0.025	0.18	0.60	0.08
TS	13.5	13.1	0.21	0.19	<0.01	0.16
Milk urea nitrogen, mg/dL	16.3	16.8	0.34	0.34	<0.01	0.93
Fat: Protein	1.38	1.30	0.051	0.30	<0.01	0.16
Somatic cell count, ×10^3^	386.6	269.9	109.74	0.46	0.89	0.99
Feed efficiency ^b^	1.51	1.63	0.011	<0.01	<0.01	0.48

^a^ ECM = Energy-corrected milk. ECM (kg) = 0.3246 × milk yield (kg) + 13.86 × fat yield (kg) + 7.04 × protein yield (kg); ^b^ Feed efficiency = milk yield/dry matter intake.

**Table 2 microorganisms-07-00562-t002:** Comparison of rumen fermentation variables in dairy cows fed *Perilla frutescens* leaf (PFL) and control (CON) diets.

Items	Treatments ^a^				SEM	*p*-Value
	CON	SD	PFL	SD		
pH	6.20	0.23	6.47	0.22	0.075	0.04
Ammonia-nitrogen, mg/dL	9.97	4.99	8.66	2.20	1.825	0.63
Concentration, mmol/L						
Total VFA ^a^	67.8	13.1	71.5	15.5	6.33	0.70
Acetate	42.6	9.56	42.8	9.34	4.12	0.96
Propionate	15.7	2.91	18.2	3.78	1.52	0.30
Butyrate	7.39	1.58	8.15	2.11	0.741	0.50
Valerate	0.99	0.22	1.15	0.28	0.099	0.29
Isobutyrate	0.41	0.09	0.43	0.12	0.052	0.79
Isovalerate	0.7	0.22	0.69	0.25	0.12	0.94
Molar proportion, %						
Acetate	62.6	3.92	59.9	1.35	1.1	0.14
Propionate	23.4	3.50	25.6	1.85	1.13	0.23
Butyrate	10.9	0.95	11.3	0.62	0.22	0.20
Valerate	1.46	0.19	1.61	0.14	0.05	0.08
Isobutyrate	0.6	0.07	0.6	0.14	0.05	0.98
Isovalerate	1.03	0.20	0.96	0.31	0.129	0.73
Acetate: propionate	2.74	0.55	2.35	0.21	0.162	0.14

^a^ VFA = volatile fatty acids; SD = standard deviations; SEM = standard error mean

**Table 3 microorganisms-07-00562-t003:** Alpha diversity indices of rumen bacteria in dairy cows fed *Perilla frutescens* leaf (PFL) and control (CON) diets.

Items	Treatments		SEM ^a^	*p*-Value
	CON	PFL		
Sobs	1147	1171	79.538	0.62
Shannon	5.12	5.58	0.429	0.13
Simpson	0.05	0.01	0.038	0.16
Ace	1379	1385	70.5	0.87
Chao	1394	1386	69.4	0.84
Coverage	0.99	0.99	0.001	0.31

^a^ SEM = standard error mean.

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
