# Peer review of "Perilla frutescens Leaf Alters the Rumen Microbial Community of Lactating Dairy Cows"

_microorganisms, 2019, doi:10.3390/microorganisms7110562_

Round 1
Reviewer 1 Report
In this study Sun et al., have evaluated the effect of Perilla frutescens leaf (PFL) on ruminal fluid microbial community in cattle. This study finds its importance towards mitigating and/or reducing methane emission, an important greenhouse gas of which ruminants have been reported as major contributors. Authors have not addressed PFL availability, accessibility or economic burden which are important deciding factors in real setting. Besides PFL medicinal benefits which are probably in human study what was the rationale behind this plant. In addition, what was the deciding factor in dosage of PFL? How efficient is the ruminal microbiome in digesting these? The study design doesn’t address questions on animal numbers, their age, diet influences before study including antibiotic treatments which may have profound impact on microbiome.
Finally, steps to identify changes in eructation methane emissions will be important in identifying whether changes in community by PFL works towards its goals.
Specific Comments:
Introduction needs more background information specifically role of rumen microbiome on growth and development of the host What other medicinal plants have been used for such purposes? Discuss a couple of studies with specific observations and results in the introduction as well. Line 52. Include information on where the animals are housed, are they part of commercial setting or university experimental setting? How many other animals are part of the farm? How were the test animals housed in location to the non-experimental animals? Please include this information in Methods as well. What was the rationale behind considering 7 animals per group? Line 59. The abstract mentions 350g/d PFL. Is that per day? Line 59 only talks about 50mg extract. How was 350grams achieved? Describe that in methods? Also explain the rationale behind administering 350g? Line 131: Is the pH of rumen normally acidic? PFL increased pH to less acid, how does this affect rumen microbial community that in turn affects health of the host? PFL treatment increased Butyrate production which has been shown to improve growth and development of host, however propionate was also found to increase as per Table 1. Propionate may have antagonist effect on the host health. How do the authors relate this to the host health? Line 141-143 is not supported by the figure. There are differences per animal but not by groups. Fig 1., does not show any visible differences and specifically for 1B no clustering is observed as well as lacking confidence ellipses. What is the purpose of this figure? Why are certain key groups in fig 4 black and others grey? Include this information in Figure legend.
Specific Minor comments:
Line 59. Elaborate on EP tube Line 60. What is the internal standard? It is difficult to link the colors to the key for both phyla as well as genera due to repeating and closely related colors. In addition to changing the colors, please include description in the legend describing main community member differences. Figure 3 should be moved to supplementary figure as it doesn’t provide additional information. The quality of the figure is poor additionally.Author Response
Reviewer 1
Comments and Suggestions for Authors
In this study Sun et al., have evaluated the effect of Perilla frutescensleaf (PFL) on ruminal fluid microbial community in cattle. This study finds its importance towards mitigating and/or reducing methane emission, an important greenhouse gas of which ruminants have been reported as major contributors. Authors have not addressed PFL availability, accessibility or economic burden which are important deciding factors in real setting. Besides PFL medicinal benefits which are probably in human study what was the rationale behind this plant. In addition, what was the deciding factor in dosage of PFL? How efficient is the ruminal microbiome in digesting these? The study design doesn’t address questions on animal numbers, their age, diet influences before study including antibiotic treatments which may have profound impact on microbiome.
AU: Thanks a lot for your comments. We totally agree with you that the animal numbers, their age, and diets may influence the efficient roles of PFL. This study is just a preliminary research towards the effects of Perilla frutescensleaf (PFL) on milk synthesis and rumen fermentation in dairy cows. To our knowledge, this is the first study that using PFL in dairy cows. For the dosage of PFL, we did a pre-trial before this experiment. A pretrial lasting for 1 week was conducted before this experiment and found that 200, 300 g/d PFL did not apparently inhibit the feed intake, but 400 g/d PFL had a tendency to decrease the feed intake. Thus, a dosage of 300 g/d PFL was selected. We add these words to the manuscript.
Finally, steps to identify changes in eructation methane emissions will be important in identifying whether changes in community by PFL works towards its goals.
AU: We did not measure the methane emissions due to the limitation of equipment. We agree with you that the detection of methane emission is important. We are going to measure methane emissions by chamber in the further related studies.
Specific Comments:
Introduction needs more background information specifically role of rumen microbiome on growth and development of the host.
What other medicinal plants have been used for such purposes?
Discuss a couple of studies with specific observations and results in the introduction as well.
AU: Thanks for your suggestion.
Previous studies found that the effect of ruminal microbiota on the host can be achieved by the short chain fatty acids released by ruminal bacteria such as Mitsokella spp(Tan et al., 2014). In addition, Shen et al. (2017) found that the effect of rumen derived short chain fatty acids on the growth and metabolism of epithelial cells was mediated by the regulatory network of G protein coupled receptor (GPR) and histone deacetylase (HDAC). Thus, the rumen microbiome has intimate connections with growth and development of the host.
It has been found that the supplement of Andrographis paniculata leaves rich in plant active substances (lactones, flavonoids, sterols) in goat diet increased the quantity of ruminal Ruminococcus albus, Ruminococcus flavefaciens, and Fibrobacter succinogenes, and then improved the nutrients digestibility (Yusuf et al., 2017).
These words were added to the Introduction.
Tan, J.; McKenzie, C.; Potamitis, M.; Thorburn, A. N.; Mackay, C. R.; Macia, L., The role of short-chain fatty acids in health and disease. Adv Immunol 2014,121, 91-119. Shen, H.; Lu, Z.; Xu, Z.; Chen, Z.; Shen, Z., Associations among dietary non-fiber carbohydrate, ruminal microbiota and epithelium G-protein-coupled receptor, and histone deacetylase regulations in goats. Microbiome 2017,5(1), 123. Yusuf, A. L.; Adeyemi, K. D.; Samsudin, A. A.; Goh, Y. M.; Alimon, A. R.; Sazili, A. Q., Effects of dietary supplementation of leaves and whole plant of Andrographis paniculata on rumen fermentation, fatty acid composition and microbiota in goats. BMC Vet Res 2017,13(1), 349.
Line 52. Include information on where the animals are housed, are they part of commercial setting or university experimental setting?
AU: It is a modern standard dairy cow barnlocated in a commercial dairy farm. The animals used in this study were housed separately with other cows using specific subfield.
How many other animals are part of the farm? How were the test animals housed in location to the non-experimental animals? Please include this information in Methods as well.
AU: The total number in this dairy farm is approximately 1500 lactating dairy cows.
What was the rationale behind considering 7 animals per group?
AU: We used 18 cows per group in the feeding experiment. And we selected 7 animals with similar milk yield that was close to the average milk yield of the group to collect rumen fluid for 16S rRNA sequencing. We rephrased the words in Animals, Diets, and Experimental Design.
Line 59. The abstract mentions 350g/d PFL.
AU: It is 300g/d PFL in the Abstract. Please check.
Is that per day? Line 59 only talks about 50mg extract. How was 350grams achieved? Describe that in methods? Also explain the rationale behind administering 350g?
AU: We are sorry for your question. This sentence is talking about the detection of main bioactive compound of PFL by LC-MS. We did this analysis using 50 mg PFL. It is not the 300 g/d used for the cows feeding.
Line 131: Is the pH of rumen normally acidic? PFL increased pH to less acid, how does this affect rumen microbial community that in turn affects health of the host? PFL treatment increased Butyrate production which has been shown to improve growth and development of host, however propionate was also found to increase as per Table 1. Propionate may have antagonist effect on the host health. How do the authors relate this to the host health?
AU: We agree with you that the pH of rumen is normally acidic. In the current study, both of the PFL (pH =6.47) and CON (6.2) groups showed the normal rumen pH value. It was reported that the common mean rumen pH value of lactating dairy cows is from 5.99 to 6.96 (Duffield et al., 2004).
No significant differences were found for the propionate and Butyrate production.
Duffield, T.; Plaizier, J. C.; Fairfield, A.; Bagg, R.; Vessie, G.; Dick, P.; Wilson, J.; Aramini, J.; McBride, B., Comparison of techniques for measurement of rumen pH in lactating dairy cows. J Dairy Sci 2004,87(1), 59-66.
Line 141-143 is not supported by the figure. There are differences per animal but not by groups. Fig 1., does not show any visible differences and specifically for 1B no clustering is observed as well as lacking confidence ellipses. What is the purpose of this figure?
AU: We reworded this sentence to “The PCoA plots (Figure 1B) showed that the clouds derived from the PFL and CON data had a tendency to be separated from each other.”
Why are certain key groups in fig 4 black and others grey? Include this information in Figure legend.
AU: We reconsidered this information in Figure 4 and followed the suggestion of Reviewer 2, we decided removed this part.
Specific Minor comments:
Line 59. Elaborate on EP tube Line 60. What is the internal standard?
AU: The internal standard is 2-Chloro-L-phenylalanine (purity ≥ 98%). We added these words.
It is difficult to link the colors to the key for both phyla as well as genera due to repeating and closely related colors. In addition to changing the colors, please include description in the legend describing main community member differences.
AU: We are sorry for the colors. We changed the colors in Phyla and genera. Please see the revised Figure 2.
Figure 3 should be moved to supplementary figure as it doesn’t provide additional information. The quality of the figure is poor additionally.
AU: We reorganized the Figure and improve the quality. Figure 3 is showing the different bacteria between the two groups using LEfSe analysis. In our view, it is very important in this study. Most of the other studies use this kind figure to show the different bacterial community among treatments.
Reviewer 2 Report
The manuscript by Sun and colleagues provides a detailed characterization of the bacterial community enriched in the rumen liquor of dairy cows when Perilla frutescens was supplied in the diet. The authors observed the enrichment of different bacteria in the control group compared to the group where P. frutescens was supplied. Furthermore, the bacterial taxa correlated to parameters linked to rumen fermentation and milk production. The manuscript is well organized and well written (however, I suggest an overall language editing, which would be useful to improve the quality). Despite a more detailed chemical characterization of parameters linked to rumen fermentation and milk (e.g. complete fatty acid profile) would have added more relevance to the results, the findings give useful insights for future studies. I have few comments that should be addressed by the authors before publication.
Main concerns.
- The authors used PICRUSt to infer the functional genes of the enriched communities. This tool exploits the 16S sequence to retrieve the genome of phylogenetically close microorganisms and could be used to infer the functional profile of a microbial community. I am always concerned when this kind of approach is used, since it assumes that phylogenetically close microorganisms have the same functional profile (i.e. same functions within the community) regardless to their origin, which is not always true. I suggest to remove this part. Alternatively, if the authors really want to report the prediction of a functional profile, I suggest to use CowPI, which was developed specifically for rumen (Wilkinson TJ, Huws SA, Edwards JE, Kingston-Smith AH, Siu-Ting K, Hughes M, Rubino F, Friedersdorff M and Creevey CJ (2018) CowPI: A Rumen Microbiome Focussed Version of the PICRUSt Functional Inference Software. Front. Microbiol. 9:1095. doi: 10.3389/fmicb.2018.01095).
- Data about milk production are used to make correlations to bacterial taxa (lines 188-194). However, I did not find a section about these analyses in the M&M. Furthermore, it could be useful to report explicitly these data in the results and to discuss them. In this version of the manuscript data are only reported in Table S2. Did the authors observed differences between the conditions?
Minor issues.
- Please check the use of italic in the title.
- A brief introduction in the abstract could be useful.
- Line 29. It is the first time that Perilla frutescens is reported in the main text. This should not be abbreviated. Abbreviations can be used later in the manuscript (e.g. line 229).
- Lines 59-64. It could be useful to add few words to make clear that this is the chemical characterization of P. frutescens.
- Lines 77-78. Please add a reference for the primers. Furthermore, was the barcode at the 5’ end of the primers?
- Lines 79-82. Where samples tested at different dilutions to check possible inhibition?
- Lines 91-98. Which was the number of sequences (total and average) before and after filtering?
- Lines 99-100. Which confidence was used for the classification?
- Lines 102-110. Where the sequences rarefied at the same value for each sample?
- line 123. There is a typo: “p- value”.
- Table 1. It is reported “P-value” (i.e. upper-case letter). Please check in the whole manuscript and be consistent.
- Line 135. Should be “Volatile fatty acids”.
- Lines 141-146. Which taxonomic rank was considered? OTUs?
- Figure 2A and 2B. There were unclassified sequences in the dataset? If yes, what is the percentage of unclassified sequences?
- Figure 2B. It is not easy to read since there are too many taxa. What about to use a cutoff higher than 1%?
- Figure 3. I like how the authors have reported the data in this figure. However, I suggest to increase the fonts and to improve quality of the figure, since it is difficult to read in this version.
- Figure 5. The fonts are too small.
- Line 246. Where members of the Archaea detected with the primer set used in this study? If yes, was their relative abundance different between the conditions?
- Lines 253-254. I think there is a jump in the contents. Maybe it could be useful to explain why the authors are discussing the presence of member of the family Muribaculaceae and then start discussing the genus Acetitomaculum.
- Line 291. The authors reported that the genus Sharpea produces lactate. Then it is reported “we compared the relative abundances of other lactate-utilizing bacteria”. I think that the content is not clear as reported.
- Lines 299-301. Please make clear that this is a speculation.
- Lines 307-309. Can the authors make a hypothesis for the possible inhibition of the genus Treponema?
- Line 308. There is typo. Is “Treponema”.
Author Response
Reviewer 2
Comments and Suggestions for Authors
The manuscript by Sun and colleagues provides a detailed characterization of the bacterial community enriched in the rumen liquor of dairy cows when Perilla frutescens was supplied in the diet. The authors observed the enrichment of different bacteria in the control group compared to the group where P. frutescens was supplied. Furthermore, the bacterial taxa correlated to parameters linked to rumen fermentation and milk production. The manuscript is well organized and well written (however, I suggest an overall language editing, which would be useful to improve the quality). Despite a more detailed chemical characterization of parameters linked to rumen fermentation and milk (e.g. complete fatty acid profile) would have added more relevance to the results, the findings give useful insights for future studies. I have few comments that should be addressed by the authors before publication.
AU: Thanks for your comments on language. For improvement of our English language, we have requested the American Journal Experts (http://www.aje.com/en) to have the English language editing. Please refer to the enclosed certificate below:
Main concerns.
- The authors used PICRUSt to infer the functional genes of the enriched communities. This tool exploits the 16S sequence to retrieve the genome of phylogenetically close microorganisms and could be used to infer the functional profile of a microbial community. I am always concerned when this kind of approach is used, since it assumes that phylogenetically close microorganisms have the same functional profile (i.e. same functions within the community) regardless to their origin, which is not always true. I suggest to remove this part. Alternatively, if the authors really want to report the prediction of a functional profile, I suggest to use CowPI, which was developed specifically for rumen (Wilkinson TJ, Huws SA, Edwards JE, Kingston-Smith AH, Siu-Ting K, Hughes M, Rubino F, Friedersdorff M and Creevey CJ (2018) CowPI: A Rumen Microbiome Focussed Version of the PICRUSt Functional Inference Software. Front. Microbiol. 9:1095. doi: 10.3389/fmicb.2018.01095).
AU: Thanks for your suggestion. We removed this part.
- Data about milk production are used to make correlations to bacterial taxa (lines 188-194). However, I did not find a section about these analyses in the M&M. Furthermore, it could be useful to report explicitly these data in the results and to discuss them. In this version of the manuscript data are only reported in Table S2. Did the authors observed differences between the conditions?
AU: Thanks for your suggestion. We added this information of milk production in the M&M, results and discussion. We changed Table S2 to Table 1 in the revised version.
Minor issues.
- Please check the use of italic in the title.
AU: Revised as you suggest.
- A brief introduction in the abstract could be useful.
AU: Revised as you suggest. “Perilla frutescens(L.) Britt., an annual herbaceous plant, has antibacterial, anti-inflammation, and antioxidant properties.” was added
- Line 29. It is the first time that Perilla frutescens is reported in the main text. This should not be abbreviated. Abbreviations can be used later in the manuscript (e.g. line 229).
AU: Revised as you suggest.
- Lines 59-64. It could be useful to add few words to make clear that this is the chemical characterization of P. frutescens.
AU: Revised as you suggest. The chemical characterization of PFL was measured by the following procedure.
- Lines 77-78. Please add a reference for the primers. Furthermore, was the barcode at the 5’ end of the primers?
AU: The reference is below.
Xu, N.; Tan, G. C.; Wang, H. Y.; Gai, X. P., Effect of biochar additions to soil on nitrogen leaching, microbial biomass and bacterial community structure. J. Soil Biol.2016, 74, 1-8.
- Lines 91-98. Which was the number of sequences (total and average) before and after filtering?
AU: The number of sequences (total and average) before filtering is 797635 and 56974, respectively. The number of sequences (total and average) before filtering is 455756 and 32554, respectively. We add this information to the 3.2 Results section in the revised version.
- Lines 79-82. Where samples tested at different dilutions to check possible inhibition?
AU: The DNA template was uniformly diluted to 10 ng for amplification. These words were added to the manuscript.
- Lines 99-100. Whichconfidence was used for the classification?
AU: The confidence used for the classification is 0.7. These words were added to the manuscript.
- Lines 102-110. Where the sequences rarefied at the same value for each sample?
AU:The sequences were rarefied at the same value (25965) for each sample. These words were added to the manuscript.
- line 123. There is a typo: “p- value”.
AU: Revised as you suggest.
- Table 1. It is reported “P-value” (i.e. upper-case letter). Please check in the whole manuscript and be consistent.
AU: Revised as you suggest. It should be p-value
- Line 135. Should be “Volatile fatty acids”.
AU: Revised as you suggest.
- Lines 141-146. Which taxonomic rank was considered? OTUs?
AU: Yes. We follow the OTUs.
- Figure 2A and 2B. There were unclassified sequences in the dataset? If yes, what is the percentage of unclassified sequences?
AU: For the genus level (Figure 2B), the percentage of unclassified sequences is 8.1% (20/247).
We add these words to the Results section.
- Figure 2B. It is not easy to read since there are too many taxa. What about to use a cutoff higher than 1%?
AU: We changed the colors of Figure 2 to make it easier to read. The cutoff is already higher than 1%.
- Figure 3. I like how the authors have reported the data in this figure. However, I suggest to increase the fonts and to improve quality of the figure, since it is difficult to read in this version.
AU: Actually, the original Figure is clear. We re-uploaded a new clear version of this Figure.
- Figure 5. The fonts are too small.
AU: The Figure 5 is revised.
- Line 246. Where members of the Archaea detected with the primer set used in this study? If yes, was their relative abundance different between the conditions?
AU: We are sorry that we did not detect the Archaea in the current study.
- Lines 253-254. I think there is a jump in the contents. Maybe it could be useful to explain why the authors are discussing the presence of member of the family Muribaculaceae and then start discussing the genus Acetitomaculum.
AU:We added the following words “Thus, Muribaculaceae has the ability to undergo acetogenesis, similar to Acetitomaculum, which belongs to the group of acetogenic bacteria,” between the two sentences.
- Line 291. The authors reported that the genus Sharpea produces lactate. Then it is reported “we compared the relative abundances of other lactate-utilizing bacteria”. I think that the content is not clear as reported.
AU: We reworded these sentences “A recent study found that the abundance of the genus Sharpea was much higher in the high milk yield and high milk protein content cows than in the low milk yield and low milk protein content cows [21], which was confirmed by the current study that the PFL cows with higher milk yield had greater abundance of ruminal Sharpea. In addition, Selenomonas that is also a lactate-utilizing bacteria [36] had greater abundance in PFL cows compared to CON cows.”
- Lines 299-301. Please make clear that this is a speculation.
AU: We agree with you that this is a speculation. Further studies will be included to confirm the functions of these bacteria in relating to milk performance.
- Lines 307-309. Can the authors make a hypothesis for the possible inhibition of the genus Treponema?
AU: It was only reported that several plant extracts had the inhibition roles of peptidase and giycosidase activities of Treponema denticola (Homer et al., 1992). Based on the previous studies, we hypothesized that the inhibition of Treponema might be due to the specific bioactive compounds in PFL.
Homer, K. A.; Manji, F.; Beighton, D., Inhibition of peptidase and glycosidase activities of Porphyromonas gingivalis, Bacteroides intermedius and Treponema denticola by plant extracts. J Clin Periodontol 1992,19(5), 305-10.
- Line 308. There is typo. Is “Treponema”.
AU: Revised as you suggest.

Round 2
Reviewer 1 Report
None